# Antioxidant Activity and Cell Protection of Glycosylated Products in Different Reducing Sugar Duck Liver Protein Systems

**DOI:** 10.3390/foods12030540

**Published:** 2023-01-26

**Authors:** Feili Zhan, Jiafeng Luo, Yangying Sun, Yangyang Hu, Xiankang Fan, Daodong Pan

**Affiliations:** 1Key Laboratory of Animal Protein Food Processing Technology of Zhejiang Province, College of Food and Pharmaceutical Sciences, Ningbo University, Ningbo 315211, China; 2State Key Laboratory for Managing Biotic and Chemical Threats to the Quality and Safety of Agro-Products, Ningbo University, Ningbo 315211, China

**Keywords:** duck liver protein, glycosylation, antioxidant properties, cellular protection

## Abstract

Duck liver is an important by-product of duck food. In this study, we investigated the effects of glucose, fructose, and xylose on the antioxidant properties of glycosylated products of duck liver protein and their protective effects on HepG2 cells. The results show that the glycosylation products of the three duck liver proteins (DLP-G, DLP-F, and DLP-X) all exhibit strong antioxidant activity; among three groups, DLP-X shows the strongest ability to scavenge DPPH, ·OH free radicals, and ABTS^+^ free radicals. The glycosylated products of duck liver protein are not toxic to HepG2 cells and significantly increase the activity of antioxidant enzymes such as SOD, CAT, and GSH-Px in HepG2 cells at the concentration of 2.0 g/L, reducing oxidative stress damage of cells (*p* < 0.05). DLP-X has a better effect in reducing oxidative damage and increasing cellular activity in HepG2 cells than DLP-G and DLP-F (*p* < 0.05). In this study, the duck liver protein glycosylated products by glucose, fructose, and xylose were named as DLP-G, DLP-F, and DLP-X, respectively.

## 1. Introduction

Duck liver accounts for 2% to 2.5% of duck weight, which is a major by-product during duck processing [1]. At present, the development and utilization of duck liver is mainly concentrated on the rough processing of food and the addition of animal feed. Therefore, the overall utilization rate of duck liver protein is low. Duck liver protein has strong functional properties such as antioxidants and cell damage repair [2]. The active peptides can be obtained from the duck liver, which can be widely used in healthy products and cosmetics because of their antioxidant, anti-inflammatory, and immune-enhancing properties. The modification products of duck liver protein can be applied in food processing for the development of functional foods. The production of global livestock and poultry is about 2365 × 10^9^ kg [3]. A small amount of duck liver is consumed by people; most is processed into animal feed or directly discarded [4]. By 2050, the demand for protein of animal origin will increase as the world’s population continues to grow [5]. Therefore, it is necessary to improve the comprehensive utilization of duck liver by-products through deep processing.

Physical, chemical, or enzymatic methods have been widely applied in protein structure modification [6]. However, there are some downsides to the chemical and enzymatic methods, when considering the high costs and potential health risks. The Maillard reaction is easier to control and fewer harmful substances are produced than chemical and enzymatic methods [7]. The essence of glycosylation is the Maillard reaction. It is a complex reaction, including the carbon group in the reducing sugar covalently bonded with the amino group in the amino acid [8]. Proteins and carbohydrates could be covalent bonded together, which leads to a better control on the conditions of the reaction [9]. The glycosylation of protein is an environmental friendly and effective method [10]. Glycosylation modification can not only effectively improve the functional properties of solubility, thermal stability, and emulsification, but also endow proteins with antioxidant ability [11]. In recent years, there was a wealth of studies into improving protein function by the methods of glycosylation. Cheng et al. [12] found dextran, coupled with rice protein, could significantly improve the protein solubility and further enhance the industrial use of rice protein. Wu et al. [13] reduced the allergenicity of turbot parvalbumin by glycosylate. Chang et al. [14] found that the glycosylated products of glucose and maize protein can be prepared into a new nano carrier for lutein delivery.

There are many factors that affect the glycosylated reaction, mainly including reaction temperature, reaction time, pH, reducing sugar type, and substrate concentration, etc. [15]. Among these factors, the influence of sugar types on the Maillard reaction has been the focus of research in recent years, including the degree of polymerization of sugars and sugar configurations [16]. Zhang et al. [17] investigated the effect of different types of reducing sugars (glucose, fructose, xylose, and lactose) on emulsification properties of forest frog ovalbumin and glycosylation products of forest frog ovalbumin, and show that the glycosylated product of xylose and egg protein has a better emulsification ability. Wang et al. [18] found that the neutral polysaccharide may be an ideal material for glycosylation, because it may provide a strong spatial position against intermolecular aggregation. Wei et al. [19] obtained gelatin hydrolysate from fish scales and then studied the effects of anti-oxidation properties of reaction products by different types of reducing sugars. The results show that the ribose has a good antioxidant capacity and the Maillard reaction products can enhance the flavor of caramel mouthfeel. Therefore, the type of reducing sugar has an important effect on the functional properties of glycosylated protein products. Glucose, fructose, and xylose were widely used in practical production and processing owing to their inexpensive character.

Duck liver is one of the important by-products during the duck industry. The traditional crude processing would reduce the economic efficiency and cause some environmental pollution. In order to improve the overall utilization of duck liver, we modified duck liver protein by glycosylation. In this study, we investigated the effects of duck liver protein glycosylated product, by different species’ reducing sugars (glucose, fructose, and xylose), on antioxidant characteristics and cellular oxidative damage protection. This study may provide a theoretical basis for improving the comprehensive utilization of duck liver by-products.

## 2. Materials and Methods

The raw material (duck liver) for the experiment was purchased from Hua Ying Food Co., Ltd. in Nanjing. Some analytically pure chemical reagents such as 1, 1-diphenyl-2-picryl-hydrazyl (DPPH), salicylic acid, glucose, fructose, and xylose were purchased from Biological Co., Ltd., Shanghai, China. Some test kits such as T-AOC, MDA, CAT, SOD, and GSH-Px test kits were purchased from Jian-Cheng Technology Co., Ltd., Nanjing, China. 2.2-BIS (3-ethyl-benzothiazoline-6-sulphonic acid), ABTS, and ethanol were purchased from Sinopharm Group Pharmaceutical Co., Hong Kong. Human hepatoma cell HepG2 was derived from the laboratory of Food and Drug College of Ningbo University in Zhejiang Province. Medium (high glucose DMEM), FBS, concentration 0.25%, was purchased from Gibco Co., Waltham, MA, USA. Some experimental consumables such as PBS, cell plate, and culture flask were purchased from Corning Co.,Nanjing, China. The CCK-8 kit was purchased from the Tohito Institute of Chemistry in Japan. The water used in this study was ultra-pure, and used the Molecular Water Purification System (Kuehler Scientific Instruments Co., Ltd., Shanghai, China) All chemical reagents used in this experiment were analytically pure. Chemical reagents were prepared according to the standards and instructions.

### 2.1. Preparation of Duck Liver Protein

The extraction method we used for duck liver protein was referred from Lu et al. [20], with minor modifications. The environment temperature for the extraction of duck liver protein should always be controlled in the range of 273.15 K to 277.15 K. Then, the duck liver was chopped after removing the sinew of fresh duck liver and pure water was added at a volume ratio of 1:6 duck liver to water. It was homogenized for 10 min at 10,000 r/min, then adjusted the pH to 11 and mixed well by a magnetic stirrer (Lichen Instrument Co., Ltd., Zhejiang, China) for 10 min. Then, it was centrifuged at 10,000 r/min for 15 min by 5804R high-speed refrigerated centrifuge (Eppendorf Company, Hamburg, Germany) and adjusted its pH in the range of 4.5~4.7 and homogenized for 10 min. The above centrifugal operation was repeated once. At this time, duck liver protein could be obtained in the precipitated state. A little amount of pure water was added to the duck liver protein in the precipitated state and centrifuged 3~4 times according to the above centrifugal conditions. In this way excess salt can be removed. Finally, duck liver protein was obtained after freeze-drying with a Christ freeze dryer (Bo Lixing Instrument Co., Ltd., Beijing, China), and its purity was 76.2%.

### 2.2. Preparation on Glycosylated Products of Duck Liver Protein

The glycosylated reaction of duck liver protein was prepared according to the method of Chen et al. [21] with some modifications. Duck liver protein was dissolved and sugars (glucose, fructose, and xylose) reduced in a beaker at the mass ratio of 1:4. The concentrations of duck liver protein and reducing sugar in the system were 1% and 4%, respectively. Then, it was magnetically stirred for 1 h and adjusted the pH of solution to 9.0 after being fully dissolved. Then, the mixture was put between duck liver protein and reducing sugar into the HH-8 digital constant temperature water bath (Lichen Bangxi Instrument Technology Co., Ltd., Shanghai, China) for 4 h at 363.15 K. It is worth noting that it should be sealed with plastic wrap to prevent the evaporation of the solution during the reaction. Finally, the glycosylated product of duck liver protein could be obtained when the thermal reaction was ended and cold. It should be noted that the reaction conditions were the optimal conditions for the glycosylated reaction of duck liver protein based on our previous research. The glycosylated product is sealed and stored at 277.15 K after lyophilized. In this study, the duck liver protein glycosylated products by glucose, fructose, and xylose were named as DLP-G, DLP-F, and DLP-X, respectively.

### 2.3. pH Measurement

The method of pH determination of Vhangani et al. [22] was emulated. The pH value was measured by the portable pH meter (Mettler-Toledo Instruments Co., Ltd., Shanghai, China). pH meter was calibrated with buffer solutions prior to use. Then the pH of the reaction solution at the end of the reaction was measured.

### 2.4. Determination of Browning Value

The method for determining browning values was based on Lu et al. [20] with minor modifications. The supernatant of the reaction solution was taken and diluted 10 times. Then homogenized by homogenizer (Staufen, Hamburg, Germany) at 5000 rpm for 30 s and measured the absorbance value of the sample at 420 nm by using UV-3300 ultraviolet spectrophotometer (MAPADA company, Shanghai, China), which can effectively evaluate the degree of browning.

### 2.5. Determination of DPPH Free Radical Scavenging Rate

The determination method of DPPH free radical scavenging rate was based on Vhangani et al. [22] with minor modifications. The DPPH was dissolved in anhydrous ethanol and the DPPH ethanol solution prepared at a concentration of 0.2 × 10^−3^ mol/L. A total of 1 mL of sample extract was taken and added to the DPPH ethanol solution (1 mL, 0.2 × 10^−3^ mol/L), then mixed well and incubated for 30 min at room temperature away from light. The absorbance value of the sample was measured at 517 nm. The 95% ethanol solution was used as a control and 0.1 g/L VC solution was used as reference. The formula for calculating the DPPH radical scavenging rate is as follows:(1)$$\mathrm{Scavenging}\;\mathrm{rate}\;\mathrm{of}\;\mathrm{DPPH}\;\mathrm{radicals}\left(\%\right)=\left[1-\frac{\mathrm{Ai}-\mathrm{Aj}}{\mathrm{Ac}}\right]*100$$
where “Ai” is the absorbance of the DPPH solution to which the sample was added. “Aj” is the absorbance of ethanol solution blank group at 517 nm. “Ac” is the absorbance value of the DPPH solution without the sample solution.

### 2.6. Determination of ·OH Free Radical Scavenging Rate

The determination method of OH free radical scavenging rate was based on Vhangani et al. [22] with minor modifications. A total of 1 mL of the sample liquid was placed in a test tube and 1 mL of FeSO_4_ solution at a concentration of 10 × 10^−3^ mol/L, 1 mL of salicylic acid–ethanol solution at a concentration of 10 × 10^−3^ mol/L, and 1 mL of H_2_O_2_ solution (0.03%, *w*/*v*) were added in turn. They were shaken well and then kept in a constant temperature water bath at 310.15 K for 30 min. Centrifuged at 6500 r/min for 5 min and measured supernatant absorption at 510 nm. The 0.5 g/L VC solution was used as reference. The formula for the ·OH free radical scavenging rate is as follows:(2)$$\cdot \mathrm{OH}\;\mathrm{free}\;\mathrm{radical}\;\mathrm{scavenging}\;\mathrm{rate}\left(\%\right)=\left[1-\frac{\mathrm{As}-\mathrm{Ao}}{\mathrm{Ac}-\mathrm{Ao}}\right]*100$$
where “Ac” is the absorbance of the sample without scavenger at 510 nm; “As” is the absorbance of the sample with scavenger at 510 nm; “Ao” is the absorbance of the blank group after replacing the sample with pure water.

### 2.7. Determination of ABTS^+^ Free Radical Scavenging Rate

The determination method of ABTS^+^ free radical scavenging rate was based on Bonvehí et al. [23] with minor modifications. A total of 0.10 × 10^−3^ kg of DLP-G, DLP-F, and DLP-X was weighed into the 15 mL centrifuge tube separately and 10 mL of distilled water was added and mixed well to make the glycosylated product at a concentration of 10 g/L. The concentration of sample diluent DLP-G, DLP-F, and DLP-X was diluted to 4.0 g/L. The ABTS solution (7 × 10^−3^ mol/L) and K_2_S_2_O_8_ solution (2.45 × 10^−3^ mol/L) were mixed at a ratio of 1:1 and then kept away from light for 12 to 16 h. ABTS radical solution was diluted by anhydrous ethanol until the absorbance of 0.70 (±0.02) at 734 nm. Then, 10 mL of sample dilution was added to 990 mL of ABTS^+^ working solution, shook well, and incubated for 6 min at 303.15 K. We measured the absorbance at 734 nm. The formula for determining ABTS^+^ free radical scavenging rate is as follows:(3)$$\mathrm{ABTS}\;\mathrm{free}\;\mathrm{radical}\;\mathrm{scavenging}\;\mathrm{rate}(\%)=\left[1-\frac{\mathrm{As}-\mathrm{Ac}}{\mathrm{Ao}}\right]*100$$
where “As” is the absorbance of the sample with ABTS radical solution at 734 nm, “Ac” is the absorbance of the sample with distilled water at 734 nm, and “Ao” is the absorbance of distilled water with ABTS radical solution at 734 nm.

### 2.8. Determination of the Total Antioxidant Capacity

The method of sample solution was prepared as the same as above. Total antioxidant capacity was determined according to the relevant instructions of the T-AOC assay kit.

### 2.9. Recovery of HepG2 Cells

At first, we incubated the tubes with HepG2 cells at 310.15 K in water bath for 3 min. Then poured in 1 mL of DMEM complete medium (containing 10% FBS, 1% double antibody) and centrifuged at 1000 r/min for 10 min. Secondly, the supernatant was removed from the centrifuge tube and 2 mL of medium added. Then it was blown slowly, transferred to a cell culture plate, 10 mL of complete medium added, and incubated in a constant temperature incubator for 48 h (310.15 K, 5% CO_2_).

### 2.10. Passaging Culture of HepG2 Cells

The HepG2 cells were digested with a concentration of 0.25% trypsin for 2 min when they reached 90% or more fusion. HepG2 cells were subculture at 1:2. In this study, we selected the logarithmic growth phase cells for testing.

### 2.11. Cryopreservation of HepG2 Cells

HepG2 cells, which were in good dispersion, were digested with trypsin (1 mL, 0.25%) for 3 min, then 1ml complete medium was added and centrifuged at 1500 r/min for 10 min. The liquid supernatant was removed immediately, then the cryoprotective solution at 1 mL was added and mixed for 10 s. They were transferred to the cryopreservative tube and placed in a gradient cooling bath. Then, they were frozen in a tank of liquid nitrogen for 5 min and removed. Finally, the HepG2 cells were stored in a refrigerator at 193.15 K for storage.

### 2.12. Cytotoxicity Tests

The determination method of cytotoxicity tests was based on Hsu et al. [24] with minor modifications. The HepG2 cells were digested by 0.25% trypsin for 2 min when they reached 90% or more fusion. They were subculture at 1:2. We selected the logarithmic growth phase cells for experiment. The 100 mL of HepG2 cell suspension was at a density of 2 × 10^5^ cells/mL per well into a 96 well culture plate and incubated at inoculated 310.15 K for 12 h. The original culture medium was discarded after the cells adhered and stabilized, and then 100 mL of cell culture medium containing duck liver protein glycosylated products was added. At this time, the final glycosylated products concentrations were 0.5, 1.0, 2.0, 2.5, 5.0, and 10.0 g/L. The cell viability was determined by method of CCK-8 after culture for 4 h. Each experimental group was set up with 5 replicates.

### 2.13. Establishment of H_2_O_2_-Induced Oxidative Damage Model of HepG2 Cells

A total of 100 mL of HepG2 cell suspension was inoculated at a density of 2 × 10^5^ cells/mL per well into a 96 well culture plate and incubated at 310.15 K for 12 h. Then 100 mL H_2_O_2_-containing incomplete medium (without serum) was added after the cells adhered and stabilized. At this time, the mass concentration of H_2_O_2_ in each well was 0 × 10^−3^, 0.6 × 10^−3^, 0.7 × 10^−3^, 0.8 × 10^−3^, 0.9 × 10^−3^ 1.0 × 10^−3^, 1.1 × 10^−3^, 1.2 × 10^−3^, 1.3 × 10^−3^, and 1.4 × 10^−3^ mol/L. The cell viability was determined by the method of CCK-8 after culture for 4 h, which was based on Hsu et al. [24] with minor modifications. Each experimental group was set up with 5 replicates.

### 2.14. Protective Effect of Glycosylation Products on H_2_O_2_-Induced Oxidative Damage in HepG2 Cells

HepG2 cells were treated with a certain concentration of glycosylation products for 24 h, when they cultured and adhered to the wall. Then, they were treated with a half-lethal dose of H_2_O_2_ for 4 h.

### 2.15. Determination of Oxidative Stress Factors (MDA, SOD, GSH-Px, CAT Activity)

Each cell group was collected and cultured as the same as the method of cryopreservation of HepG2 Cells. They were centrifuged for 10 min at 1000 r/min and washed 3 times with PBS. The cells were disrupted by JY92-IIN Ultrasonic Cell Disruptor (Xinzhi Co., Ltd., Ningbo, China) at 400 W. The cells were placed in an ice-water bath during the ultrasonic treatment process. The cells were ultrasonically treated every 8 s with an interval for 10 s and the cumulative ultrasonic treatment was 8 min.

### 2.16. Statistical Analysis

Analysis of variance (ANOVA) was used to evaluate the differences between samples by SPSS software (Version 27, IBM Inc., Armonk, NY, USA). The test for the data is Duncan’s multiple-range tests and the statistical differences were significant at α = 0.05 level. Histogram analysis was performed by HIPLOP software (Bioinformatics open source community https://hiplot.com.cn, accessed on 10 September 2022). Each experiment was repeated three times.

## 3. Results and Discussion

### 3.1. Changes in pH

The effects of glycosylated product with glucose, fructose, and xylose on pH are shown in Figure 1A. The pH of the system is significantly lower than that of control after the Maillard reaction (*p* < 0.05). The pH of DLP-F is significantly lower than DLP-G and higher than DLP-X (*p* < 0.05). The results show that the fructose and xylose can significantly reduce the pH of glycosylation reaction compared to control (*p* < 0.05). The pH of the system drops from 9 to 5.5 after the Maillard reaction. The pH of xylose reaction system is the lowest, which may be owing to reacting more readily than hexose [25]. Liu et al. [26] found that the consumption of amino groups and the formation of organic acids may be the reason for the decrease in pH in the Maillard reaction system. Therefore, it could be concluded that the types of sugar could significantly affect the pH of the reaction. The pH of DLP-X is significantly lower than that of DLP-G and DLP-F (*p* < 0.05).

### 3.2. The Effects on the Browning Value

The degree of glycosylated reaction could be effectively reflected by measuring the degree of browning of the reaction system [27]. The effects of glycosylated product with glucose, fructose, and xylose on browning value are shown in Figure 1B. The results show that the reducing sugars could increase the browning value of the glycosylated reaction. The browning value in the reaction system of fructose and xylose is significantly higher than that of glucose (*p* < 0.05). The phenomenon may be owing to fact that there are some differences in the content of α-DCs in glucose, fructose, and xylose, which are the key intermediates of Maillard reaction and the significant precursors of color [28]. Different types of reducing sugars in different resulting structures may have a different reaction. Chen et al. [16] found that the reactivity of xylose is higher than that of glucose. Measuring the absorbance of the reaction solution at 294 nm could reflect the amount of the intermediate product [29]. The intermediate product contributes less to the color but more to the Maillard reaction because it is an important precursor to the Maillard reaction [30].

### 3.3. DPPH Radical Scavenging

Glycosylated reaction of duck liver protein could scavenge DPPH free radicals by providing hydrogen atoms to form stable DPPH molecules, and the color of the reaction system changed from purple to yellow. The determination of DPPH free radical scavenging is the standard for evaluating antioxidant properties [31]. From Figure 1C, we see that the ability to scavenge DPPH free radicals is significantly improved compared to the control after the glycosylated reaction of duck liver protein (*p* < 0.05). The free radical scavenging rate of DPPH in the control is 40.7%. The DPPH free radical scavenging rate of DLP-G, DLP-F, and DLP-X is 76.76%, 84.10%, and 90.74% respectively, which is 1.87, 2.07, and 2.23 times that of the control group, respectively. This may be because the glycosylated modification forms a good hydrogen donor, which can effectively scavenge DPPH free radicals [32]. Yu et al. [33] reported that the type of sugar had an important influence on the antioxidant properties of Maillard reaction products. The Maillard reaction products using ribose has better antioxidant properties. The results show that the glycosylated products of duck liver protein using glucose, fructose, and xylose all have a good ability to scavenge DPPH free radicals, among which, GLP-X has the best effect of scavenging DPPH free radicals.

### 3.4. The Removal Rate of ·OH

·OH has strong biological activity in oxygen free radicals, which can cause damage to adjacent biomolecules. Rao et al. [34] found that the scavenging ability for ·OH can be improved by the Maillard reaction. The ability of the Maillard reaction products to scavenge hydroxyl radicals may relate to the metal chelating properties [25]. From Figure 1D, we see the effects on glycosylated products of duck liver protein using glucose, fructose, and xylose on the scavenging rate of hydroxyl radicals. The hydroxyl radical scavenging rate of duck liver protein in the control is 45.85%. The antioxidant capacity of duck liver protein glycosylation products is significantly improved compared with the control (*p* < 0.05). Among them, DLP-F and DLP-X have the strongest scavenging ability for hydroxyl radicals and the scavenging rates of ·>OH are 90.66% and 93.02%, respectively, which is close to 0.5 g/L VC (98.90%). In recent years, some studies have shown that the ability of duck liver protein glycosylation products to scavenge hydroxyl radicals may relate to the type of sugar and the degree of Maillard reaction [35]. In general, the ability of GLP-F and GLP-X to scavenge hydroxyl radicals is significantly higher than that of GLP-G (*p* < 0.05).

### 3.5. ABTS^+^ Free Radical Scavenging Rate

ABTS^+^ has strong solubility in lipids and aqueous solutions, so it is widely used in lipid-soluble and water-soluble natural substances to improve the antioxidative activity [36]. As can be seen from Figure 1E, the ability to clear ABTS^+^ of glycosylated products of duck liver protein is significantly enhanced compared to control (*p* < 0.05). There is no significant difference in the clearing ability of ABTS^+^ between GLP-G and GLP-F (*p* > 0.05). GLP-X has the strongest ability in clearing ABTS^+^. The clearance rate of ABTS^+^ is over 80%. Similar results have been obtained in recent studies. Hayase et al. [37] reported that the glycosylated products using xylose glycine exhibited stronger ABTS^+^ free radical scavenging activity. The enhanced scavenging ability of ABTS^+^ at the primary stage of the Maillard reaction may be because of the good hydrogen donors provided by intermediates and browning products. Wang et al. [38] reported that the glycosylated reaction may lead to a change in reducing sugars in the structure and endow glycosylated products with reducing ability. In general, the increasing of intermediate and end products on the Maillard reaction will help to improve the scavenging ability of ABTS^+^ [39].

### 3.6. Total Antioxidant Capacity

Fe^3+^ can be reduced to Fe^2+^ by antioxidants under the acidic conditions, thus, the total antioxidant capacity of glycosylated products can be converted by the equivalents of Fe^2+^. From Figure 1F, we can see that the total antioxidant activity of glycosylated products of duck liver protein is significantly increased compared to control (*p* < 0.05). The overall antioxidant activity of the substances is GLP-X > GLP-F > GLP-G, moving from strong to weak. Previous studies reported the activity of reducing sugars: xylose > aldose > hexose > disaccharide [40]. It can be concluded that the glycosylated products using different kinds of reducing sugars would have different antioxidants. Studies in recent years also reached similar conclusions. Wang et al. [38] found that the Maillard reaction products between protein and xylose were a good electron donor. At the same time, the electron supplied by it can reduce Fe^3+^ to Fe^2+^ and react with free radicals. Jia et al. [41] studied the function and stability of glycosylation products using xylose and rice protein. The results show that the glycosylation products has high levels of antioxidants and stability.

### 3.7. Cytotoxicity Assay

In order to establish an effective model on cellular oxidative damage, we need to determine the suitable concentration of duck liver protein glycosylation products by cytotoxicity assay. The effects of glycosylated products of duck liver protein using glucose, fructose, and xylose on the survival of HepG2 cells are shown in Table 1. The results show that the survival rate of HepG2 cells is significantly reduced (*p* < 0.05) when the concentration of glycosylated products is in the range of 0.5 to 10 g/L. Worthy of attention is the fact that the cell survival rate of the DLP-X group is only 22.67%, suggesting that glycosylated products of duck liver protein using xylose at high concentrations may have a toxic effect. The glycosylated products of duck liver protein using three kinds of reducing sugars have no significant effect on cell proliferation (*p* > 0.05). They all achieve a survival rate over 90% when the concentration is below 2.0 g/L. The control survival is 98.03%. Therefore, the glycosylated products concentration was selected at 2.0 g/L in the cell experiments.

Oxidative stress is mainly manifested in the imbalance of antioxidant systems, which is caused by the accumulation of reactive oxygen species in the body. Oxidative stress ultimately leads to oxidative damage or cell death [42]. H_2_O_2_ could easily penetrate cell membranes and produce highly toxic ·OH free radicals, which leads to the oxidative stress response of cellular lipid. At the same time, the membrane is eroded by free radicals and its permeability and integrity are damaged, eventually leading to cell damage [43]. H_2_O_2_ is often used to construct oxidative stress damage models [44]. Figure 2 shows the effects of H_2_O_2_ with different concentrations, with incubation for 4 h, on the survival rate of HepG2 cells. The survival rate of HepG2 cells gradually decreases when the final concentration of H_2_O_2_ increases. The survival rate of cells is over 80% when the concentration of H_2_O_2_ is less than 0.7 × 10^−3^ mol/L, and that is reduced to 50.55% when the concentration is 1.0 × 10^−3^ mol/L. The viability of HepG2 cells is significantly reduced to 10.24% (*p* < 0.05) when the concentration is higher than 1.0 × 10^−3^ mol/L. The cell survival should be maintained in the range of 50% to 70% in the establishment of the cell oxidative damage model [45]. We need to consider that the oxidative damage is not obvious if the cell survival rate is too high, but it is easy to cause irreversible damage if the survival rate is too low. Therefore, the concentration of H_2_O_2_ was selected at 1.0 × 10^−3^ mol/L to construct the HepG2 cell injury model in this study.

### 3.8. Protective Effect of Glycosylated Proteins on Oxidative Damage of Cells

As can be seen from Figure 3A, the glycosylated products of duck liver protein have obvious protective effects on HepG2 cells when the concentration is 2 g/L. The cell viability of the normal group is up to 99% and the model group is only 50%, which is significantly lower than normal group (*p* < 0.05). GLP-X has the best effect on improving cell vitality, which could restore 85% of the normal group, followed by GLP-F, which could restore 75% of the normal group. The reason for this phenomenon may relate to the structure of the sugar base donor [46]. The above results show that the glycosylated products of duck liver protein can effectively inhibit hydrogen-peroxide-induced cell apoptosis. Different glycosylated donors have important effects on the inhibition effect of hydrogen-peroxide-induced cell apoptosis. The antioxidant activity of GLP-X is better than GLP-G. Chailangka et al. [47] found that the functional properties of the Maillard reaction products for cricket protein were significantly different with different types of reducing sugars.

Automatic oxidation of lipids is a chain reaction triggered by free radicals, including chain initiation, chain transfer, and chain termination [48]. The content of MDA could accurately reflect the degree of lipid oxidation and cellular oxidative stress. The higher the content of MDA, the higher the degree of cellular oxidative stress and the higher the cellular oxidative damage caused [49]. Some endogenous antioxidant enzymes exist in the antioxidant system of cells, such as GSH-Px, SOD, and CAT, which could reduce cellular oxidative damage by controlling the dynamic balance of free radicals in the body. For example, the GSH-Px enzyme in cells can promote the reaction of H_2_O_2_ with GSH to produce water and oxidize the glutathione (GSSG), thereby reducing the oxidative damage in cells caused by H_2_O_2_. The enzyme of SOD could remove super-oxide anion radical and prevent oxidative damage of cells [50].

It can be seen from Figure 3 that there are significant changes in the oxidation factors of HepG2 cells after oxidative damage by H_2_O_2_. The cell activity of the model group significantly decreases and the MDA content significantly increases when compared with the normal group (*p* < 0.05). Interestingly, SOD activity, CAT activity, and GSH-Px activity of model group decreases significantly (*p* < 0.05), indicating that the lipid and antioxidant enzyme system of HepG2 cells have significantly changed and the cells have reached oxidative stress state.

Oxidative damage of these cells is reversible. The cell activity, SOD activity, CAT activity, and GSH-Px activity are significantly increased (*p* < 0.05) and the content of MDA significantly decrease (*p* < 0.05) after treatment with glycosylated products of duck liver for 24 h compared with the normal group. DLP-X can significantly reduce the MDA production (*p* < 0.05), indicating that it has a good effect on inhibiting the lipid peroxidation. The results show that glycosylated products of duck liver can up-regulate the activity of SOD, CAT, and GSH-Px antioxidant enzymes and maintain the balance of the redox-active in cells. Thereby, it can reduce the oxidative damage that is caused by H_2_O_2_ and improve the cell survival rate. Some studies found that different glycosylated products had certain differences in the expression of key proteins, some of which could regulate cell apoptosis and, ultimately, lead to oxidative damage in cells [51].

## 4. Conclusions

The glycosylated products of duck liver protein using glucose, fructose, and xylose all show good antioxidant properties and have strong scavenging ability of DPPH, -OH free radicals, and ABTS^+^ free radicals. The glycosylated products can effectively protect HepG2 cells and significantly increase its activity when the concentration is at 2.0 g/L. They can inhibit the lipid oxidation by promoting the activity of SOD, CAT, GSH-Px, and other antioxidant enzymes and reduce the content of MDA, thereby reducing the extent of cellular oxidative damage. Therefore, we can conclude that the glycosylated products of duck liver protein using glucose, fructose, and xylose all have good antioxidant properties and good cytoprotective properties, among which the glycosylated products of duck liver protein using xylose have the best antioxidant effect in vitro.

## Figures and Tables

**Figure 1 foods-12-00540-f001:**
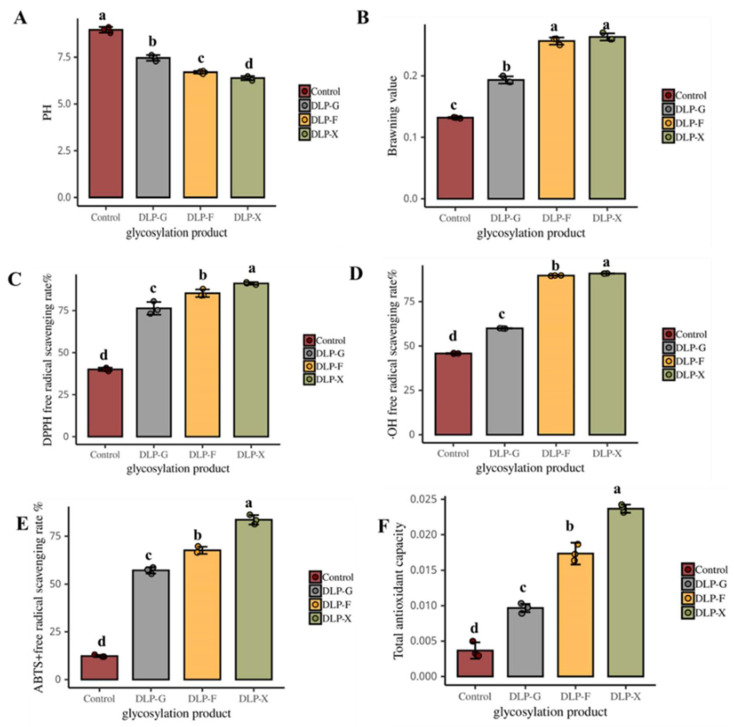
Effect of different reducing sugars on the in vitro antioxidant properties of duck liver protein glycosylation products they should be listed as: (**A**) shows the effect of different reducing sugars on the pH of the duck liver protein glycosylation products; (**B**) shows the effect of different reducing sugars on the degree of browning of the duck liver protein glycosylation products; (**C**) represents the effect of different reducing sugars on the scavenging of DPPH radicals by duck liver protein glycosylation products; (**D**) represents the effect of different reducing sugars on the scavenging of -OH radicals by duck liver protein glycosylation products; (**E**) represents the effect of different reducing sugars on the scavenging of ABTS^+^ radicals by duck liver protein glycosylation products; (**F**) represents the effect of different reducing sugars on the total antioxidant capacity of duck liver protein glycosylation products. (Note: Different letters indicate that the samples are significantly different at α = 0.05 level. The duck liver protein glycosylated products by glucose, fructose, and xylose were named as DLP-G, DLP-F, and DLP-X, respectively).

**Figure 2 foods-12-00540-f002:**
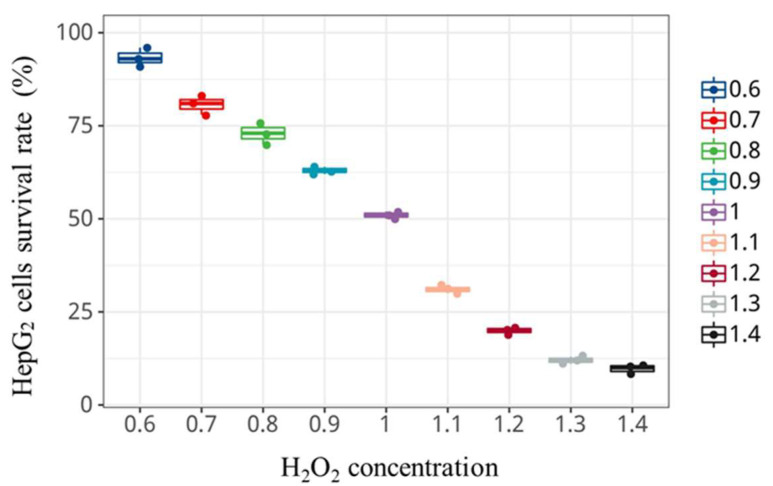
The effect of H_2_O_2_ concentration on the survival rate of HepG2 cells.

**Figure 3 foods-12-00540-f003:**
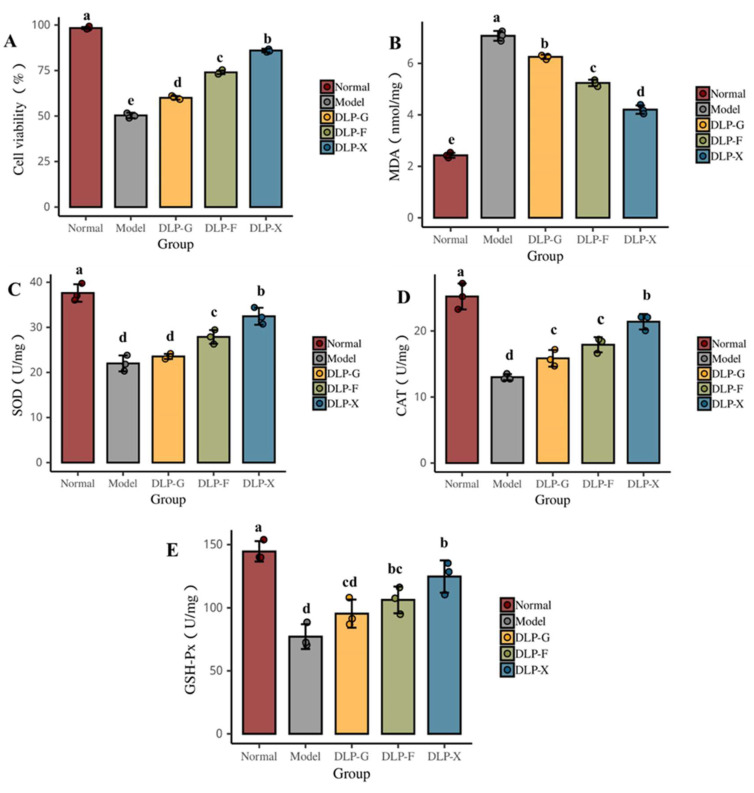
Effect of different glycosyl donor duck liver protein glycosylation products on protection against oxidative damage in HepG2 cells. (**A**) shows the effect of the glycosylated products of duck liver protein using glucose, fructose, and xylose on the survival rate of HepG2 cells, while (**B**–**E**) show the effect of the glycosylated products of duck liver protein using glucose, fructose, and xylose on the activity of oxidative stress factors MDA, SOD, CAT, and GSH-Px in HepG2 cells, respectively) (Note: Different letters indicate that the samples are significantly different at α = 0.05 level).

**Table 1 foods-12-00540-t001:** Effect of duck liver protein glycosylation products concentration on the survival rate of HepG2 cells.

Concentration (g/L)	Reducing Sugar Type		
	DLP-G	DLP-F	DLP-X
0.5	112.9 ± 11.04 ^a^	110.54 ± 13.23 ^a^	125.76 ± 4.54 ^a^
1	104.68 ± 8.54 ^ab^	105.75 ± 5.60 ^a^	115.68 ± 13.32 ^b^
1.5	98.67 ± 5.42 ^b^	96.27 ± 1.25 ^b^	93.67 ± 1.81 ^c^
2	95.86 ± 3.76 ^b^	94.12 ± 4.60 ^b^	90.84 ± 3.72 ^c^
2.5	97.87 ± 4.09 ^b^	79.83 ± 9.18 ^c^	81.93 ± 3.85 ^d^
5	82.32 ± 8.87 ^c^	71.13 ± 8.75 ^c^	55.44 ± 5.41 ^e^
10	57.20 ± 7.49 ^d^	32.75 ± 4.12 ^d^	22.67 ± 0.77 ^f^

Data results were calculated using mean ± standard deviation. Different letters in the same column indicated significant differences at α = 0.05 level.

## Data Availability

The data presented in this study are available on request from the corresponding author.

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
