# Peer review of "Antioxidant Activity and Cell Protection of Glycosylated Products in Different Reducing Sugar Duck Liver Protein Systems"

_foods, 2023, doi:10.3390/foods12030540_

Round 1

Reviewer 1 Report

Effects of different reducing sugars on antioxidant properties of duck liver protein glycosylation products and protection against cellular oxidative damage

The research work focus on the effects of different reducing sugars on antioxidant properties of duck liver protein and its protection against cellular oxidative damage.

Due to poor usage of English, and lot of grammatical mistakes, many parts of this paper are difficult to comprehend. I think authors should opt for any language service and resubmit this paper for review.

Some general comments are:

Title of the paper can be more concise. Abstract should be more focused.

Line 38-39: rephrase it.

Line 50-52: rephrase it

Line 52 & 53: Give space after full stop

Line 58: Citation style is different. It should be Hui Yun et al. [18]

Line 60, 61, and 63: Citation style is different. Same as above. Also check author name.

Line 70: Citation style is different.

Line 147: Citation style is different

Line 188: no space between 37 and ℃. It should be 37℃.

Line 203: placed in a liquid nitrogen tank. Storage. ????? What is this???

·         In materials and methods section, No reference of the methods are quoted. Preparation of duck liver protein, preparation of duck glycosylation by following ???????? method???? pH was determined by following ???????? method? Browning value by ??????  method??

·         Somewhere minutes has been used and at some places “min” was mentioned. Format should be the same throughout the text

·         Too many mistakes in the references

Reviewer 2 Report

It is desirable to reduce the introduction section

In section 3.1 Changes of pH, reference should be made to Figure 1 or a table of pH data should be added

Spectral analysis of Maillard reaction should be inserted

Future use of duck liver should be described (where it will be used, in what form, etc.)
